# Equalizing the Playing Field and Improving School Food Literacy Programs Through the Eyes of Teens: A Grounded Theory Analysis Using a Gender and Sport Participation Lens

**DOI:** 10.3390/nu17040685

**Published:** 2025-02-14

**Authors:** Alysha L. Deslippe, Coralie Bergeron, Olivia Y. Wu, Tamara R. Cohen

**Affiliations:** 1Human Nutrition, Faculty of Land and Food Systems, University of British Columbia, Vancouver, BC V6T 1Z4, Canada; alysha.deslippe@bcchr.ubc.ca (A.L.D.); olivia.wu@tufts.edu (O.Y.W.); 2Healthy Starts, British Columbia Children’s Hospital Research Institute, Vancouver, BC V5Z 4H4, Canada; coralie.bergeron@ubc.ca; 3Women and Children’s Health Sciences, Faculty of Medicine, University of British Columbia, Vancouver, BC V6T 1Z3, Canada; 4Food, Nutrition and Health, Human Nutrition, Faculty of Land and Food Systems, University of British Columbia, Vancouver, BC V6T 1Z4, Canada

**Keywords:** dietary behaviours, athletes, food literacy, gender, adolescence

## Abstract

**Background**: School food literacy programs (e.g., home economics) are an opportunity to improve the dietary habits of teens. However, the literature suggests that girls and athletes have better food literacy, and it is not clear how school programs contribute to this inequality. To address this, we explored how gender and sport influenced teens’ perspectives of their school food literacy experiences and how they can be improved. **Methods**: Using semi-structured interviews and a Grounded Theory analysis, we generated a theoretical understanding of how to improve school food literacy programs for athletes and non-athletes of diverse genders. Thirty-three teens were recruited to balance sport participation (n = 18 athletes) and gender (n = 15 boys; n = 14 girls; n = 4 non-binary) based on data saturation. **Results**: Teens expressed four categories to improve school programs that aligned with principles of the Capability, Opportunity, Motivation and Behaviours (COM-B) Model of behaviour change. Programs should *Provide a challenge* (e.g., more advanced recipes), *Make it fun* (e.g., explore new cuisines in interactive ways) and *Establish importance* (e.g., health impacts). *Practice is key* for teens’ self-confidence and development of food skills (e.g., meal planning) as well. Boys emphasized *Make it fun* whereas girls and non-binary teens emphasized *Establishing importance*. Athletes valued *Practice is key* more than non-athletes. **Conclusions**: School programs should relay the importance of food literacy in fun and tailored ways to teens (e.g., meal planning among athletes). It may be especially salient for programs to tailor their activities and messaging, where possible, to appeal to diverse teens who play sports and those who do not.

## 1. Introduction

School food literacy programs, such as home economics classes, represent a key opportunity to improve teens (13–18 years) eating behaviours [1]. Food literacy encompasses a range of skills or competencies such as knowledge on how to cook, what is considered healthy, and where food comes from [1,2,3,4,5]. Knowledge of the role of food in culture and global sustainability, as well as the interconnections between economic and physical food systems, are also important food literacy competencies [1,3]. A review of 11 countries that examined food education policies revealed that across countries there is no standardized approach or consensus in food literacy programs in terms of content or how they are delivered [6]. The other global literature has supported the notion that food literacy programs in schools do not have a set curriculum or standard implementation strategy [7,8]. Despite this, it is well supported that food literacy programs in schools are an important intervention strategy to improve the health of youth globally [6,7].

In a systematic review of 44 studies of school-based food literacy programs from 16 countries, the authors found that participation in a school food literacy program had benefits on youths’ (10–19 years) dietary habits (e.g., improved consumption of fruits, vegetables and dairy) and knowledge of food-related skills (e.g., cooking safety and nutrition knowledge) [9]. However, little research has sought to understand who specifically benefits from school food literacy programs and why. As girls have consistently been found to possess greater food literacy competencies compared with boys in several countries, including Australia, Iran, South Africa and the United States [9], and girls have been found to have greater opportunities to learn skills in countries such as Canada [10] and China [11], boys may be at a disadvantage in current school food literacy systems.

Historically, gender norms postulate that mothers are the primary parent in charge of food-related tasks [12,13,14]. This may contribute to greater pressure and opportunities for girls to learn food literacy compared with boys [9,10,11]. In a study exploring the food-related attitudes of 836 teens (11–18 years), cooking skills were viewed as more relevant for girls compared with boys [11]. Over a decade later, these findings are still evident; in a qualitative study that explored changes in dietary behaviours, the authors reported that parents expected and encouraged their daughters, but not their sons, to learn food-related skills, such as how to cook [10].

To this end, gender norms exist when discussing household division of labour, including tasks such as cooking, and they may impart a view among teens that only certain teens need to learn food literacy skills [12]. Additionally, parents may promote these norms unintentionally by providing differing opportunities for their sons and daughters to learn food literacy skills at home [10]. This inequality in the pressures and opportunities boys and girls have to learn food literacy skills may simultaneously disadvantage both groups in different ways. For example, girls may be exposed to more traditional views of ‘food-related’ work as they grow up, unknowingly creating greater opportunities for them to participate in food-related activities at home or schools [10]. This may seem positive as greater food literacy has been correlated with beneficial dietary habits [9], but gender-based pressures may also create greater stress among teen girls to master food literacy in comparison with their peers who are boys. In contrast, boys may not be provided with as many opportunities to learn about food literacy [10] or they may avoid participation altogether if they feel that food literacy skills are ‘feminine’ [15], potentially creating a negative impact on dietary habits and long-term physical health.

The literature has also suggested that sport participation may impact food literacy. In certain situations, young athletes may receive tailored information from a sports dietitian, an expert in nutrition, about how to support their physical performance through food [16,17]. This information often focuses on sport nutrition (e.g., knowledge of elevated nutrient needs and timing meals) to optimize sports performance [18], a subset of food literacy. [2]. In the literature, few studies have evaluated athletes’ food literacy [19]. Instead, most of the literature focuses on a specific subset of sport nutrition, such as safe supplement use [16,17,20], or is conducted in elite sport settings [16,17].

In studies assessing teen athletes knowledge of sport supplementation, athlete boys express greater knowledge of protein supplementation compared with girls and are better able to outline proteins role in performance [16,17,19]. In contrast, athlete girls have been found to report greater knowledge of vitamins and minerals for general health [16,17]. These trends may be influenced by historical gender norms as dietary protein has long been tied to masculinity [21,22], whereas micronutrients have long been associated with femininity [23]. As such, athlete boys and girls may pursue different types of sport nutrition information or experience different opportunities to hear about the impact of certain nutrients based on these historical connotations. This may have unfavourable consequences on an athletes’ health and performance that vary by gender; athlete boys may not pursue nutrition information about general health whereas athlete girls may not pursue information more specific to the performance context and muscle building. However, not all of the literature has found that gender plays a significant role in athletes’ sport nutrition knowledge. In a study from the United States that evaluated 535 high school athletes (14–18 years), the authors found that knowledge about sports nutrition in general (e.g., “*Importance of diet*”) did not differ between boys and girls [19]. As food literacy is a complex topic that includes various aspects such as cooking skills and nutrition knowledge among other factors [2], there is a need to clarify how sport involvement and gender impact school food literacy program experiences.

Understanding how gender and sport involvement impacts teens’ food literacy experiences is important as it has the potential to refine their design. For example, understanding why an athlete or a non-athlete may choose to participate in a food literacy program, and what they take away from the program, can point to gaps in current program delivery or content that can be amended to better address teens’ motivation to participate. At present, nothing in the literature to our knowledge has sought to untangle these differences. As motivation plays a significant role in the likelihood of behaviour change, this is an important gap to fill [24,25]. To provide guidance that food literacy program developers can use to refine current programs, we generated a theoretical understanding of how to improve teens’ experiences and engagement in school food literacy programs. As there are currently no standardized programs in terms of content or delivery within food literacy programs, this work has the potential to improve the appeal of food literacy programs where athletes and non-athletes may both be present.

## 2. Materials and Methods

This study is part of a broader project called the EATing in a Gendered world study (EatGen). The EatGen study is a mixed-methods project that aims to co-design a school-based intervention to improve high school athletes’ eating habits. This specific study is a part of the foundational ‘basic behavioural science’ phase [26] that seeks to understand the key issues and differences in athletes’ and non-athletes eating habits and experiences surrounding food. The full interview guide used in the broader study can be found published here [27,28]. In brief, the guide was developed by the first and senior authors using the Socio-Ecological Model [29], the Food Literacy Competencies for Young Adults Framework [2] and previous research [10]. The guide was pilot tested with one teen athlete prior to its use and minor changes to wording were made to improve clarity. Data were collected until saturation [30]. The checklist for consolidated criteria for reporting qualitative research (COREQ) can be found in the Appendix A.

### 2.1. Design

We recruited teens who attended a local secondary school (British Columbia, Canada) using a scannable QR code that was placed on posters around the school or shown following classroom presentations. Three female research assistants conducted the in-person presentations. Recruitment occurred at two time points; a flow diagram detailing this can be found in Figure 1. Interested teens left their contact information through the QR codes and received access to an electronic copy of the study’s consent or assent form that explained the study’s purpose and procedures, as well as the researchers’ reasons for conducting the study. All contacted teens who met the eligibility criteria were provided with a letter for their parents. To be scheduled for a one-on-one interview, a teen had to complete a brief online survey asking about their demographics (e.g., age, grade, sex, gender and ethnicity) and sport history collected on Qualtrics (Qualtrics^XM^., Provo, UT, USA, 2020). Once completed, an online interview was scheduled while the teen was in a private location via Zoom (Zoom Video Communications Inc., San Jose, CA, USA, 2016). Interviews lasted 19–57 min and were conducted by a female research assistants (one of the three who led recruitment) who had been trained by the study’s first author. The audio from the interviews was stored on a secure server with case notes. All audio was sent to a transcription service that uses humans to transcribe the data verbatim into de-identified word documents (Transcription Heroes, Inc., Toronto, ON, Canada, 2023).

### 2.2. Setting and Participants

To be eligible, a teen had to have no diagnosed eating disorder or severe dietary restriction (e.g., Crohns disease), have access to Zoom and be a Canadian citizen. Teens could be in any grade (grades 8 to 12) and were recruited to balance sport participation and biological sex. Teens who had participated in a competitive high school sport and/or a club sport in the previous year were considered ‘athletes.’ These sports compete outside of school hours several nights a week and include competition against other local schools or clubs. The content and design of food literacy programs at the chosen school are regulated at the school district level and can be found online (https://curriculum.gov.bc.ca/curriculum/continuous-views accessed on 10 December 2024). The curriculum focuses on helping teens understand the role of food advertisements in eating, how to cook meals that do not have a high degree of difficulty and the relationships between food and health. How or what content is used to achieve this is not mandated by the school or the province and instead left up to the individual teacher.

### 2.3. Analysis

Informed by Grounded Theory [30,31], we explored how gender and sport influenced teens’ food literacy experiences to develop a theoretical understanding of how to improve school programs. As best-practice methods in dietary intervention development suggest the use of a behavioural theory and framework to guide design-related decisions [32], a Grounded Theory analysis was considered appropriate to begin developing such a theoretical framework in the context of food literacy programs. The purpose of a Grounded Theory analysis is to construct a theory from data, making it well suited to this aim [30,31]. All transcripts were independently coded by the first author and either the second or third author using NVivo 12 (QSR International Pty Ltd., Burlington, MA, USA, 2025) [33]. The three researchers analyzing the data were females with various backgrounds in sport. Triangulation between the three was used to resolved any discrepancies [34]. We developed an inductive coding scheme (i.e., codes were generated from the data itself) through the process of line-by-line coding through multiple passes of each transcript [30,31]. We then conducted a process of focused coding, where similar codes or codes exploring related concepts were grouped together to form categories [30,31]. Categories were defined and their relationships to one another were explored to ensure all the ideas that teens expressed were encompassed without overlap between categories. Throughout this process, memo-writing was used to capture notes on developing codes and interconnections between categories [30,31]. Categories from transcripts were then consolidated into higher-level categories to inform a theoretical understanding of how to improve school programs [30]. The naming of high-level categories was inductively derived using teens’ wording itself. We then compared higher-level categories to constructs from established behavioural theories, such as the Capabilities, Opportunities, Motivation and Behaviours (COM-B) Model [35], Self-Determination Theory (SDT) [36] and Social Cognitive Theory (SCT) [37] that are commonly used in dietary interventions. We elected to explore this mapping as food literacy programs are ultimately a dietary-based intervention and, as such, would benefit from using guiding theory in their design [32]. By mapping inductive categories against known theories, we were able to see if there was better alignment towards using one over another, or if something completely novel was needed to guide the refinement of food literacy programs. As data were collected at two time points, we used concurrent data collection strategies and stopped data collection once theory saturation had been achieved (i.e., no new categories emerged) [30,31]. Categories were also explored based on sport involvement and gender.

## 3. Results

We conducted interviews with 33 teens. The majority of the teens were white and in grade 10 (Table 1). Just over half were considered athletes (55%), and of these most athletes participated in basketball and volleyball. All teens had participated in at least one school food literacy program previously. Four categories captured the teens’ views on how food literacy programs could be improved, as follows: *Provide a challenge*, *Establish importance, Make it fun*, and *Practice is key* (Figure 2). Though these categories captured all the teens’ ideas, gender and sport participation impacted how and why different teens prioritized different categories. Each category will be presented and discussed by differences based on sport involvement and gender.

### 3.1. Provide a Challenge

Teens expressed that food literacy programs in schools should be challenging to them. This could be achieved by providing greater depth in the content shared, such as how nutrients impact the body, or by learning how to cook more complex foods.

“I want to learn a little bit more in depth about food and… what you should have in your diet or what you need every single day… breaking down what’s inside of it.”Participant 28, girl, non-athlete

“I will probably just more so want to learn how to cook, like more complex meals.”Participant 27, girl, non-athlete

In many cases, the idea of an impending challenge built up excitement or anticipation for what was to come in food literacy classes.

“[I’d like to learn] how to do fish and like seafood in general. In my foods class towards the end of the year we were supposed to do sushi and some of the seafood stuff. However, we ran out of time in the class, and they weren’t able to cover it [and I had wanted to].”Participant 14, boy, non-athlete

Contrasting with this, some teens outlined how food literacy competencies can be hard to understand and apply (e.g., label reading) and programs needed to be simpler. Despite the challenge of learning food literacy competencies, all teens recognized that the programs were useful as many did not have the space or confidence to practice on their own.

“I would just say um, making it more, like easier to understand [would improve my experiences].”Participant 4, boy, athlete

“Do I ever make food? Only in foods class. But, not at home… I’m not too good at cooking and neither are my friends… we don’t generally cook stuff.”Participant 7, boy, athlete

*Provide a challenge* was mentioned most often by non-binary teens and girls compared with only half of boys as a useful way to improve food literacy programs. It was valued similarly between athletes and non-athletes.

### 3.2. Establishing Importance

Outlining how food impacts health and well-being were critical to promote buy-in for participating in school programs. In many cases, teens specifically talked about how this knowledge was increasingly important as they were going to be on their own at university.

“You need to know how to cook some very basic things… especially if you’re going to university… So, if you want to stay actually healthy, and not die of cancer at 50, then you got to know what to make for yourself.”Participant 3, boy, athlete

Teen girls highlighted a need to emphasize the importance of food beyond controlling body shape, whereas all athletes (regardless of gender) focused on the need to learn more about how food impacts sport performance. Almost all girls and all non-binary teens suggested *Establish importance* as their number one suggestion to improve food literacy programs. Among boys, a little over half talked about this category as a key suggestion. Most athletes also suggested this category as important.

“I think I would rather… just kind of learn more about how, like how different foods truly affect your body and how they can influence your overall being rather than just how you look.”Participant 27, girl, non-athlete

“[like to learn] Why healthy foods give you more energy than not healthy food. Like, because, like I feel like it doesn’t make sense because I feel like sugar would give you energy.”Participant 22, girl, athlete

Suggestions to emphasize the importance of food literacy included detailing how food literacy is a life skill and utilizing sources of information from national guidelines that are considered trustworthy to teens. Teens considered trusted sources as individuals possessing relevant credentials (e.g., trained teachers, nutritionists, government sources etc.), regulatory bodies (e.g., school boards, national nutrition guidelines such as Health Canada’s Canada Food Guide), or other sources perceived as being unbiased.

“Because you’ll need how to cook for the rest of your life… maybe like bringing in like a nutrition specialist and just talk about it.”Participant 1, boy, athlete

“It’s mostly all government recommendations, like the different types of food and stuff. I trust it, it makes sense to me. We have watched some documentaries though that I think are very opinionated—like vegan documentaries that I think show clips out of context. So, I don’t trust those.”Participant 9, boy, athlete

Information learned about eating to improve and manage health was often taken to heart and enacted by teens outside of the classroom.

“We were learning about content, like packaging… And so now I kinda am more cautious about what I eat. Like if I’m getting a bag of chips I’ll look at the amount of salt… if it’s more than 15% it’s probably not very good for you.”Participant 24, girl, athlete

### 3.3. Make It Fun

In addition to the importance of food on health and well-being, teens suggested that food literacy programs in schools should be fun to help motivate teens to participate. *Make it fun* was talked about as the most important suggestion to improve food literacy programs among almost all boys and non-athletes. *Make it fun* was also highlighted among all non-binary teens as a priority, tied with *establishing importance* for the most important.

“I really enjoy cooking, even before I took that class, so I knew that I would enjoy it. However, there were a lot of people in the class that were just taking it because it’s an easy A. I think really one element that could be emphasized a bit more is the fun that you could have with cooking.”Participant 14, boy, non-athlete

Specific suggestions on how to incorporate more ‘fun’ into school programs included considering what teens wanted to learn, using clear dialogue, and supportive teachers.

“I would probably make the foods that you want to make, because they sort of just like give you food you have to make, instead of like your own personal choice.”Participant 12, boy, non-athlete

“I think simplified more. Because there was a lot of stuff that I didn’t understand, there was just a lot of big words, and it was kind of boring.”Participant 24, girl, athlete

“It’s less focused on the course and more focused on the teacher. She was very negative about things… anytime people would make a mistake, she would get very aggressive about it, which I think that kind of discourages people from wanting to take foods classes.”Participant 16, non-binary, non-athlete

### 3.4. Practice Is Key

A major aspect that contributed to enjoyment in school food literacy programs was the fact that they were hands-on and provided space to practice skills such as cooking.

“I feel like everyone should have the hands-on experience… while creating the things that are the best for your body.”Participant 20, girl, athlete

This was especially critical in improving teens’ confidence and knowledge to perform cooking tasks on their own without guardian supervision.

“Making food can be pretty daunting. I used to not like it because the few experiences I’ve had before foods class it took super long and it didn’t turn out great and this class showed me that it’s a lot easier.”Participant 9, boy, athlete

“I don’t cook very much at all. I cook when there’s nothing in the fridge or there’s no food at all. And usually, it’s just for me or me and my little sister.”Participant 30, girl, non-athlete

An emphasis on practice was especially prominent among athletes who spoke about needing to understand how to perform food-related tasks on their own to manage their fuelling strategies. There was a stark difference between how non-athletes and athletes prioritized this suggestion. *Practice is key* was viewed by majority of athletes as critical to school programs whereas it was not focused on by non-athletes.

“I’m kind of just happy that I know how to cook the basic stuff so that I can always cook myself a meal when I need to so I’m never hungry when I have to go to sports.”Participant 8, boy, athlete

## 4. Discussion

Teens’ participation in food literacy programs in schools marks a key opportunity to improve their dietary habits. To understand how to improve such programs, we explored 33 teens’ opinions on what makes school food literacy programs appealing to develop an underlying theory for program refinement. Four categories captured teens ideas, including *Provide a challenge, Establish importance*, *Make it fun* and *Practice is key,* making up the theoretical underpinnings. These underpinnings aligned closely with the principles of the Capability, Opportunities, Motivation and Behaviours (COM-B) Model [35], and subsequently our findings suggest that incorporating COM-B into the development of future food literacy programs may have positive impact on teens’ dietary habits (Figure 3). Further, our findings add to the existing literature by examining how sport involvement and gender impact teens’ buy-in for school programs. Athletes emphasized the importance of *Practice is key* as food literacy competencies were viewed as a tool for sport performance. Boys were keener to suggest the importance of *Make it fun* whereas girls and non-binary teens wanted to see greater emphasis on *Establishing importance*. These differences fill a current gap in our understanding as to why diverse teens, including boys, girls, athletes and non-athletes, may participate in school food literacy program.

### 4.1. COM-B Can Guide Refinement of Food Literacy Programs in Schools

Given how close the four categories (i.e., *Provide a challenge*, *Establish importance*, *Make it fun* and *Practice is key*) identified in this analysis are to the core concepts of the COM-B model, the COM-B model may be an advantageous model to use when designing or modifying school food literacy programs. Nothing in the other literature, to our knowledge, has sought to evaluate how school food literacy programs can be improved in a systematic way using behavioural theory.

COM-B may offer specific advantages as a guiding theory over others such as Self-Determination Theory [36] or Social Cognitive Theory as it allows for the incorporation of the home environment [37] through the ‘opportunity’ component [35]. Literature from Australia supports this notion, as researchers determined that the COM-B model was an appropriate and comprehensive model to assess how the use of meal kits impacted parents’ food literacy competencies in the home [38]. As the other literature has consistently suggested that incorporating the home environment and parental support in their teens’ dietary habits is advantageous [39,40,41], the COM-B model may provide more direct inclusion of the home compared with other behaviour change theories. This may be especially salient among boys, as previous research conducted in a similar setting has suggested that boys may have less support in the home to learn about learn food literacy [10].

### 4.2. Athletes View Food and Food Literacy Competencies Differently to Non-Athletes

Large differences in how athletes and non-athletes valued learning food literacy competencies arose. Athletes drew connections with how competencies such as cooking and meal planning supported sport performance, and as a result were motivated to learn these skills. This is exemplified by the importance that athletes placed on *Practice is key*, which was minimally valued among non-athletes. Other studies have suggested that teen athletes are motivated to improve their sport performance and may seek out nutrition information for this reason, even to the extent of considering steroids [42]. The literature from the United States further found that simply self-identifying as an athlete, regardless of performance goals, corresponded with a greater intake of sports drinks such as Gatorade [43]. As such, teen athletes may be uniquely motivated to follow certain dietary strategies if they believe them to fit the habits of a ‘typical’ athlete, even if they are not evidence-based. Thus, food literacy programs may offer an opportunity to help empower teen athletes’ knowledge of evidence-based sport nutrition information by tailoring some of their content.

### 4.3. Traditional Gender Norms May Impact Teens’ Motivation to Learn About Food Literacy

Our findings show that diverse gender groups, including boys and non-binary teens, valued participating in school food literacy programs. This is a substantial difference compared with previous research conducted in high school settings; girls expressed more opportunities and desire to learn such skills [9,10,11]. While these are positive findings, suggesting that gender norms are not motivating factors for teens to learn about food literacy, our results also warrant caution. One main difference that arose in our work was that girls and non-binary teens suggested *Establishing importance* and *Provide a challenge* as the most critical aspects that improved the appeal of school programs. In contrast, boys highlighted *Make it fun*. This gender-based difference may suggest that traditional norms in food-related roles still prevail to some extent. The contrasting values of boys and girls in our sample match the historical literature exploring gender norms [23] and the division of food work in the home [14], in that girls are expected to be healthier and manage food work for a family to ensure nourishment. Other studies have found that connotations in society surrounding certain types of food work, such as barbequing or being a professional chef, emphasize the ‘fun’ side of food for men, but not for women [13,44]. As such, future research should explore how the wording in school food literacy programs can be adjusted to reinforce the importance of food literacy for health among boys without undermining the fun side of food literacy for girls.

### 4.4. Next Steps for Educators and Policy Makers in School Food Literacy Programs

Like any behaviour change intervention, school food literacy programs should be subject to rigorous development, monitoring and assessment [45]. As a first step, school policy makers and educators could administer an open-ended feedback survey among teens after a few weeks of a food literacy program using key prompts that evaluate the theoretical underpinnings of the program. Based on our findings, this survey could ask teens if the program (1) provides a challenge?; (2) is fun?; (3) clearly explains why food literacy is important for health?; and (4) provides enough time to practice the skills they are taught? Feedback from these surveys should be incorporated to refine the programs. To ensure that any changes made meet teens’ needs, teens should remain involved alongside educators [46]. A second suggestion for food literacy educators and policy makers is the need to consider providing teens greater space for self-exploration. For example, having teens prepare a short script outlining how a nutrient of their choice is important for their health or performance as an athlete. An activity of this nature would capture the unique motivators of a diverse group, and in theory could help motivate participation. Finally, policy makers in educational settings should include assessments of ‘success’ which can be meaningfully compared across food literacy programs in diverse schools and countries [47]. This would greatly increase our understanding of what a successful food literacy program should strive to achieve regardless of geographic boundaries. As schools face numerous time and resource constraints, our suggestions may not be feasible in all school settings. Greater work is needed to understand the feasibility of our findings in real-world settings.

### 4.5. Strengths and Limitations

This study is one of the first to consider how participation in sport and gender may impact food literacy experiences in schools. We employed best-practice methods such as pilot testing the interview guide with teens to ensure its appropriateness in wording. We utilized triangulation and memo-writing to help mitigate potential bias between researchers’ own experiences while analyzing the data too. Further, the researchers who coded the data had great diversity in their sport experiences to help account for potential bias based on sport involvement from impacting the analysis. Our recruited sample was also gender diverse and accounted for the opinions of teens who played a wide range of sports, though we did not consider differences in responses based on the sport played. Despite these strengths, it must be acknowledged that our sample was relatively homogenous in terms of grade and ethnicity. As such, care should be taken when generalizing the findings to other groups, especially ethnically diverse teens who are from lower socio-economic neighborhoods as our findings may not capture the unique experiences of teens from these contexts. Future work should include a more diverse sample of teens from different grades and ethnic backgrounds to improve our understandings. We also did not account for any differences among athletes in terms of how much they trained, and it is possible that differences may arise between athletes who play sports more often than those who do not. Finally, as we did not measure teens’ actual dietary behaviours, future work is needed to assess if school food literacy programs do indeed have the long-term behavioural impacts they are believed to possess.

## 5. Conclusions

Food literacy programs intended for high school teens should be tailored in ways that motivate all teens, regardless of gender or sport participation. By doing so, teens may be more likely to pursue food literacy programs and adopt health-protective dietary habits. In the classroom, food literacy educators and policy makers should consider greater inclusion of teens’ self-exploration to tailor the learning experience to their unique needs. Balancing the messaging of the importance of food literacy skills for health in addition to the role of food in personal enjoyment should further be included in the development of these programs. Finally, our results suggest that food literacy programs need critical evaluation that considers teens’ values and perspectives.

## Figures and Tables

**Figure 1 nutrients-17-00685-f001:**
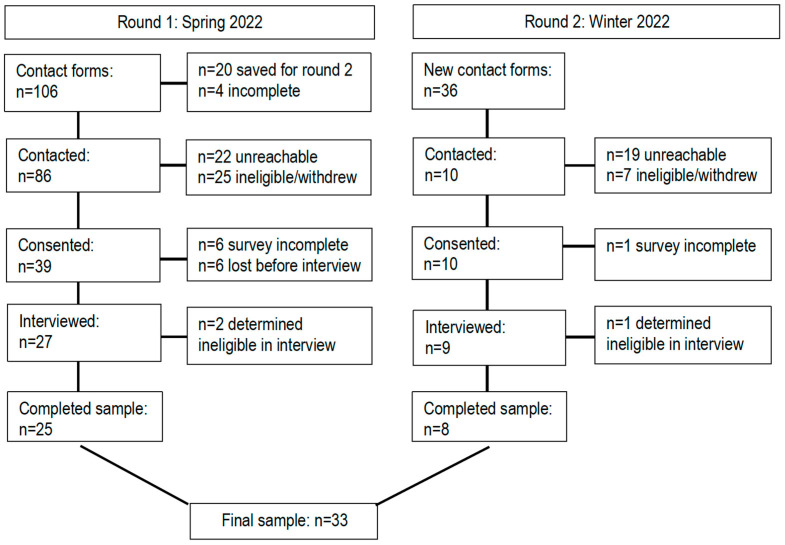
Flow diagram of participant recruitment (n = 33 teens).

**Figure 2 nutrients-17-00685-f002:**
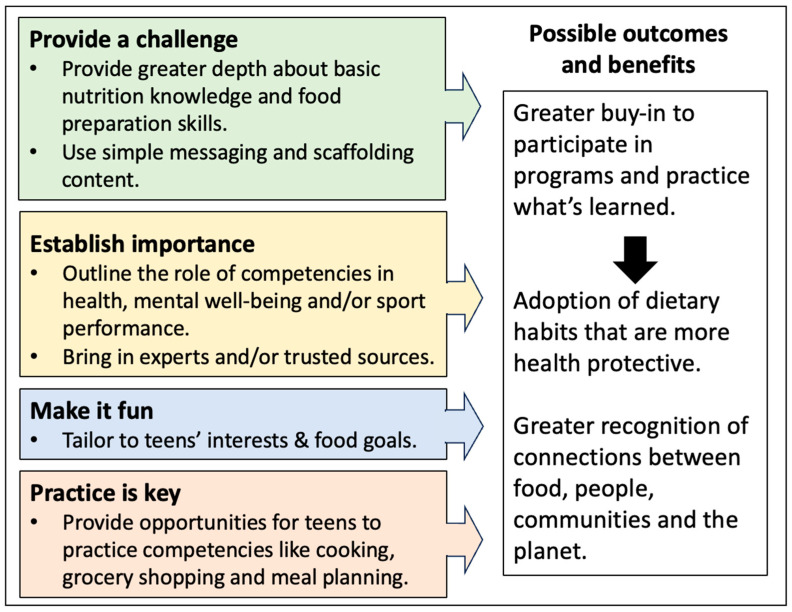
Theoretical underpinnings to improve school food literacy programs suggested by teens.

**Figure 3 nutrients-17-00685-f003:**
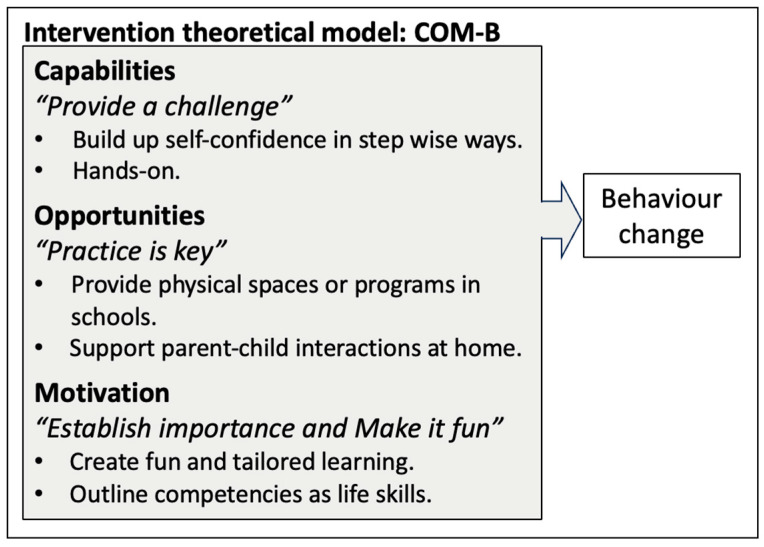
Mapping of the inductively derived theoretical underpinnings to the Capability, Opportunity, Motivation and Behaviours Model (COM-B). Categories that were identified by teens in this analysis can be found in *italics* below each COM-B core component.

**Table 1 nutrients-17-00685-t001:** Characteristics of interviewed teen athletes and non-athletes from British Columbia, Canada (n = 33).

Teen	Sex	Gender	Grade	Ethnicity	Athlete?
1	Male	Boy	10	White	Yes
2	Male	Boy	9	Mixed	Yes
3	Male	Boy	10	Mixed	Yes
4	Male	Boy	10	White	Yes
5	Male	Boy	9	White	Yes
6	Male	Boy	12	White	Yes
7	Male	Boy	10	White	Yes
8	Male	Boy	11	White	Yes
9	Male	Boy	9	White	Yes
10	Male	Boy	10	Korean	No
11	Male	Boy	9	White	No
12	Male	Boy	12	White	No
13	Male	Boy	8	Filipino	No
14	Male	Boy	12	White	No
15	Male	Boy	9	White	No
16	Male	Non-binary	12	White	No
17	Female	Girl	11	White	Yes
18	Female	Girl	10	White	Yes
19	Female	Girl	10	White	Yes
20	Female	Girl	12	White	Yes
21	Female	Girl	10	Chinese	Yes
22	Female	Girl	10	White	Yes
23	Female	Girl	9	White	Yes
24	Female	Girl	8	White	Yes
25	Female	Girl	8	Undisclosed	Yes
26	Female	Non-binary	11	White	No
27	Female	Girl	11	White	No
28	Female	Girl	10	White	No
29	Female	Girl	10	White	No
30	Female	Girl	12	Chinese	No
31	Female	Non-binary	9	Mixed	No
32	Female	Non-binary	9	White	No
33	Female	Girl	10	Hispanic	No

Gender was self-declared by all teens. Sex was based on the sex listed on a passport. n = number. Sport involvement included aquatics, basketball, baseball/softball, cross country, dance, field hockey, horseback riding, football, hockey/skating, lacrosse, rowing, rugby, soccer and volleyball. Teens often played more than one sport in school and club settings.

## Data Availability

Data available by request from the corresponding author.

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
