# Peer review of "Equalizing the Playing Field and Improving School Food Literacy Programs Through the Eyes of Teens: A Grounded Theory Analysis Using a Gender and Sport Participation Lens"

_nutrients, 2025, doi:10.3390/nu17040685_

Round 1
Reviewer 1 Report
Comments and Suggestions for Authors
The abstract is well structured and well written, the reader is able to form an opinion about the content of the manuscript. In the introduction a gap in the literature is identified. In the introduction, the gap in the literature is mentioned, but could be further detailed. For example, it would be useful to explain why it is important to look at gender differences and how these might affect the participation and benefits of food literacy programs. Also, the impact of sport participation could be explained more clearly and in more detail in the context of the existing literature.
The methodology is well structured. However, it would be helpful to have a clearer explanation of the recruitment, data analysis and use of theory to make the study more valid and transparent. For example, the definition of 'athlete' and 'non-athlete' is based on having participated in competitive school or club sport in the preceding year, which may introduce bias. This does not allow for variation in the level of participation or previous sporting experience, which could affect the results, as some young people may have more intense or recent sporting experience than others. There is mention that there has been theoretical saturation and that concurrent data collection strategies have been used, but there is no full clarification of how this has been evaluated. The results are well presented, but the discussion could be improved by making more of a comparison between your own results and those obtained in previous research. The conclusion should be rewritten. In its current format it is superficial compared to the other parts of the article (results and discussion).
Author Response
We thank the reviewer for their insightful feedback. Our responses are indicated below in italics. Changes in the manuscript are indicated with the corresponding line number.
- In the introduction a gap in the literature is identified. In the introduction, the gap in the literature is mentioned, but could be further detailed. For example, it would be useful to explain why it is important to look at gender differences and how these might affect the participation and benefits of food literacy programs. Also, the impact of sport participation could be explained more clearly and in more detail in the context of the existing literature.
We have expanded on the rationale and literature outlining how gender (line 58-80) and sport (81-17) impact food literacy experiences. Specifically, we have outlined how motivation to participate and opportunities to participate differ depending on gender and sport involvement and how this may impact a teens’ health or performance as an athlete.
- The methodology is well structured. However, it would be helpful to have a clearer explanation of the recruitment, data analysis and use of theory to make the study more valid and transparent. For example, the definition of 'athlete' and 'non-athlete' is based on having participated in competitive school or club sport in the preceding year, which may introduce bias. This does not allow for variation in the level of participation or previous sporting experience, which could affect the results, as some young people may have more intense or recent sporting experience than others.
As many teens drop out from sport in adolescence, we aimed to capture the recent experiences of teens who were actively or recently active in sports. This is important as participation 5 years ago may not reflect the context of a high school athlete and instead, reflect the experiences of a pre-teen, or a teen who retired from sports. Though important, these situations would represent a different aim then our own of understanding the experiences of current high school athletes in school food literacy programs. Within our definition athletes of diverse training levels would be captured as mentioned. This is in line with our aim to understand high school athletes’ experiences in these settings as there is a diversity in who participates from elite level athletes to teens who only participate because their peers play. To ensure our findings can be allied to athletes in all of these settings, we did not limit our criteria to a specific level of weekly training. We have added an acknowledgement of this as a limitation in terms of understanding differences based on training level (line 475-478).
- There is mention that there has been theoretical saturation and that concurrent data collection strategies have been used, but there is no full clarification of how this has been evaluated.
We have clarified what theory saturation meant in the context of our analysis in line 198. We stopped recruitment once no new categories arose in the analysis process.
- The results are well presented, but the discussion could be improved by making more of a comparison between your own results and those obtained in previous research.
We have added additional references to the literature throughout the discussion and compared them to our findings. (line 398-400, 411-413, 434-436).
- The conclusion should be rewritten. In its current format it is superficial compared to the other parts of the article (results and discussion).
We have edited the conclusion to align more clearly with our main findings and the discussion sections.
Reviewer 2 Report
Comments and Suggestions for Authors
The manuscript addresses a relevant and timely topic, providing a solid qualitative analysis of adolescents’ experiences with school food literacy programs, with a focus on gender and sports participation. The structure is clear and coherent; however, there are different areas where improvements can enhance the overall quality of the work.
1. Originality and Relevance of the Topic
The topic is highly relevant to the scientific and educational communities. However, the manuscript could further emphasize its originality compared to existing literature. The authors could strengthen the comparison with previous studies by highlighting the specific research gap that this study aims to fill. Regarding this aspect, the authors could consider the following paper:
Moscatelli et al., Assessment of Lifestyle, Eating Habits and the Effect of Nutritional Education among Undergraduate Students in Southern Italy., Nutrients. 2023 Jun 26;15(13):2894. doi: 10.3390/nu15132894.
2. Clarity of Objectives
The study objectives are well outlined, but the introduction could better clarify how the findings can inform the design of future food literacy programs. The authors could add a paragraph explicitly linking the expected results with practical implications for educational policy and program development.
3. Methodology
The use of grounded theory for qualitative analysis is appropriate; however, the description of data analysis could be more comprehensive.
Suggestions:
- Provide a more detailed explanation of the coding process, including how key themes were identified and refined.
- Discuss potential researcher biases and how they were mitigated.
- Offer a clearer justification for choosing grounded theory over other qualitative approaches.
4. Sampling and Representativeness
The participant recruitment process is well described, but the limited ethnic diversity in the sample may affect the generalizability of the findings. Discuss the implications of the sample composition in greater depth and suggest how future research could address this limitation.
5. Discussion and Practical Implications
The discussion provides a good link to theoretical frameworks but could be more concise and focused. Here are some suggestions for the authors to improve the discussions:
- Strengthen the discussion on gender differences in the findings and their implications for inclusive program design.
- Include more concrete recommendations for educators and policymakers.
- Address potential barriers to the implementation of the study’s suggestions in real-world school settings.
Author Response
We thank the reviewer for their insightful feedback. Our responses are indicated below in italics. Changes in the manuscript are indicated with the corresponding line number.
- The topic is highly relevant to the scientific and educational communities. However, the manuscript could further emphasize its originality compared to existing literature. The authors could strengthen the comparison with previous studies by highlighting the specific research gap that this study aims to fill. Regarding this aspect, the authors could consider the following paper:
We have added a few lines that re-enforce the originality of our work (line 113-114 and line 379-380). We appreciate the paper suggestion and will keep it in mind for a future publication. Give our sample, we want to ensure the literature we present focuses on teens.
- The study objectives are well outlined, but the introduction could better clarify how the findings can inform the design of future food literacy programs. The authors could add a paragraph explicitly linking the expected results with practical implications for educational policy and program development.
We have expanded on this in the introduction in lines 107-112.
“Understanding how gender and sport involvement impacts teens’ food literacy experiences is important as it has the potential to refine their design. For example, understanding why an athlete or a non-athlete may choose to participate in a food literacy program, and what they take away from the program, can point to gaps in current program delivery or content that can be amended to better meet teens’ motivation to participate”
- The use of grounded theory for qualitative analysis is appropriate; however, the description of data analysis could be more comprehensive.
Suggestions:
- Provide a more detailed explanation of the coding process, including how key themes were identified and refined.
- Discuss potential researcher biases and how they were mitigated.
- Offer a clearer justification for choosing grounded theory over other qualitative approaches.
We have added in additional details as to how we generated codes inductively in line 172 and how categories were compared to one another (line 179). In line 182-183 and 187-188 we clarify our process with additional details on how higher level category were development and how we mapping to known behavioral theories.
In qualitative research, triangulation and memo-writing are common practices to help mitigate bias in the data analysis process. We further had a diversity in our coders sport experiences to help account for bias based on participation of the researchers themselves. We have elaborated on these strengths regarding bias mitigation in the discussion (lines 463-466)
As the aim of our analysis is to inform the development of future food literacy programs and best practice methods for intervention design involve use of a guiding theoretical framework, grounded theory is the most appropriate choice for our aim. Other qualitative techniques are incredibly useful, but do not construct a theory from data. As such, using a grounded theory analysis means that we meet our aim to develop a guiding theory that can be used to refine food literacy programs. We have added text in the methods to explain this more clearly (line 171-175).
- The participant recruitment process is well described, but the limited ethnic diversity in the sample may affect the generalizability of the findings. Discuss the implications of the sample composition in greater depth and suggest how future research could address this limitation.
We have discussed this as a major limitation of our study in the discussion and included a call for future research. We have modified the text to add greater depth. (line 471-475).
- The discussion provides a good link to theoretical frameworks but could be more concise and focused. Here are some suggestions for the authors to improve the discussions:
- Strengthen the discussion on gender differences in the findings and their implications for inclusive program design.
- Include more concrete recommendations for educators and policymakers.
- Address potential barriers to the implementation of the study’s suggestions in real-world school settings.
Only one main gender difference arose in our findings. We have altered wording in (line 425) to make this clearer and outlined next steps that are needed to clarify what should be done to address this (436-438).
We have renamed paragraph 5 (line 439) in the discussion to Next steps for educators and policy makers in school food literacy programs”. We have amended wording throughout this section to make the calls to action here clearer and expanded on our examples (lines 440-457).
We have highlighted time and resource constraints that schools face in implementing the strategies we have suggested more clearly in lines 457-459.